# Necessary conditions for sustainable water and sanitation service delivery in schools: A systematic review

**Christine JiaRui Pu**[1]*, **Poojan Patel**[1], **Gracie Hornsby**[1], **Gary L. Darmstadt**[2], **Jennifer Davis**[1,3]

**1** Department of Civil and Environmental Engineering, Stanford University School of Engineering, Stanford, California, United States of America, **2** Department of Pediatrics, Stanford University School of Medicine, Stanford, California, United States of America, **3** Woods Institute for the Environment, Stanford University, Stanford, California, United States of America

* cjpu@stanford.edu

## Abstract

Access to water, sanitation, and hygiene (WASH) services confers significant health and economic benefits, especially for children, but only if those services can be delivered on a consistent basis. The challenge of sustainable, school-based WASH service delivery has been widely documented, particularly in resource-constrained contexts. We conducted a systematic review of published research that identifies drivers of, or tests solutions to, this challenge within low- and middle-income countries (PROSPERO 2020 CRD42020199163). Authors in the first group employ cross-sectional research designs and interrogate previously implemented school WASH interventions. Most conclude that dysfunctional accountability and information sharing mechanisms drive school WASH service delivery failures. By contrast, most of the interventions developed and tested experimentally by authors in the second group focus on increasing the financial and material resources available to schools for WASH service delivery. Overall, these authors find negligible impact of such infusions of cash, infrastructure, and supplies across a variety of sustainability outcome metrics. Taken together, the evidence suggests that sustainable service delivery depends on three simultaneously necessary components: resources, information, and accountability. Drawing upon theory and evidence from social psychology, public management, and political science, we identify priority knowledge gaps that can meaningfully improve the design of effective interventions. We also highlight the importance of both interdisciplinary collaboration and local expertise in designing WASH programming that aligns with sociocultural and institutional norms, and is thus more likely to generate sustainable impact.

## Introduction

Schools are important settings for water, sanitation, and hygiene (WASH) investments. Students spend a significant portion of their time at schools, are more vulnerable to infection than adults, and can transmit pathogens from the school to the home environment [1–3]. Access to

**Data Availability Statement:** All relevant data are within the paper and its Supporting information files.

**Funding:** This study was supported in part by World Vision (25799, awarded to J.D., https://www.worldvision.org/), USAID funding through WASHPaLS (WASHPaLS-G05-Stanford, awarded to G.D. and J.D., https://www.globalwaters.org/washpals), the Natural Sciences and Engineering Research Council of Canada (PGSD3 – 532687 – 2019, awarded to C.P., https://www.nserc-crsng.gc.ca/index_eng.asp), and the Scampavia Family gift to Dr. Gary Darmstadt at the Stanford University School of Medicine. The funders had no role in study design, data collection and analysis, decision to publish, or preparation of the manuscript.

**Competing interests:** The authors have declared that no competing interests exist.

school WASH services—defined here as the reliable availability of safe drinking water, clean and functional sanitation facilities, and handwashing infrastructure with soap and water—has been documented to reduce the incidence of enteric diseases, respiratory infections, and absenteeism in students [4]. These benefits are contingent on the routine practice of healthy WASH behaviors, which in turn depend on the sustainable delivery of WASH services. At a minimum, sustainable WASH service delivery includes two components: (1) regular maintenance and repair of physical infrastructure and (2) consistent provision of necessary consumables (e.g., soap and water).

Empirical evidence suggests that the sustainable delivery of WASH services in schools is widely lacking, particularly in resource-constrained contexts. These challenges manifest, for example, in poorly maintained infrastructure and inconsistent provision of WASH consumables [5–8]. Limited prior research has posited that key reasons for unsustainable service delivery include inadequate budgets and resources, unreliable water sources, irregular monitoring and oversight, and ineffective accountability mechanisms [9–12]. Recent interventions that have targeted these barriers, singly or in combination, have had mixed results at best [13–20].

We conducted a systematic review of literature focused on the sustainability of WASH services in schools, with the high-level goal of identifying necessary and sufficient conditions for sustainable service delivery. More specifically, we draw on two decades of literature to: (1) synthesize evidence regarding enablers of and barriers to sustainable service delivery; (2) examine the extent to which, and the conditions under which, previously implemented interventions have succeeded in delivering sustainable services; and (3) identify remaining knowledge gaps and priorities for future research.

## Methods

### Protocol registration and sample frame

This review was registered with Prospero (CRD42020199163, see S1 Protocol). We followed PRISMA guidelines for reporting (see S1 Checklist).

Studies were eligible for inclusion if they focused on the maintenance of WASH infrastructure and/or the provision of WASH consumables in schools. WASH infrastructure was defined to include any engineered facility that provided drinking water, sanitation, or handwashing services. Maintenance was defined as any activity that kept the WASH infrastructure clean and functional, either preventatively or in response to repair needs. Schools were defined as public or private educational institutions that catered to students up to grade 12.

Peer-reviewed studies published in English and after the year 1999 were eligible for inclusion. Studies using any research design were included, provided they were conducted in a school setting in a low- or middle-income country (LMIC). No other geographic or population restrictions were applied.

### Search strategy and study selection

A set of seminal papers on WASH service delivery in schools (n = 9) was identified from the authors' prior knowledge of the literature. Keywords were extracted from the titles and abstracts of these papers and used to create a search string. The search string was iterated on until it returned all 9 seminal papers in SCOPUS, Web of Science, and PubMed.

In May 2020, an electronic search was conducted in English across the three selected databases. The search string combined key terms using the following structure: [school] AND [WASH] AND [sustainability]. The full search string was: (CHILD* OR HEADMASTER* OR KID* OR PUPIL* OR SCHOOL* OR STAFF OR STUDENT* OR TEACHER*) AND (BATHROOM* OR CONSUMABLE* OR FACILITIES OR FACILITY OR HAND PUMP*

OR HAND WASH\* OR HANDPUMP\* OR HANDWASH\* OR HYGIEN\* OR INFRA-STRUCTURE\* OR LATRINE\* OR SANIT\* OR SOAP\* OR TOILET\* OR WASH OR WATER OR WATER CLOSET) AND (ACCESS\* OR ACCOUNT\* OR BARRIER\* OR BUD-GET\* OR CHAMPION\* OR CLEAN\* OR CONDITION\* OR CONTINU\* OR DELIVER\* OR ENABL\* OR ENVIRONMENT\* OR EXTERNAL\* OR FEASIB\* OR FUNCTION\* OR GOVERN\* OR IMPROV\* OR INSPECT\* OR INSTITUTION\* OR MAINT\* OR MANAGE\* OR MONITOR\* OR O&M OR OPERAT\* OR POOR\* OR PROVI\* OR QUALIT\* OR RECURR\* OR REGULAT\* OR RELIAB\* OR REPAIR\* OR SERVICE\* OR STRATEG\* OR SUPPL\* OR SUSTAIN\* OR USABILITY OR USABLE OR USE OR WELL\*).

Email alerts from all search engines were created and monitored on a weekly basis from May 2020 to July 2021 to capture papers published after the initial search.

Following deduplication of the initial search results, two researchers independently screened the titles and abstracts of all remaining publications for eligibility using a standard protocol developed by the authors. The two researchers piloted this protocol on a set of 21 papers to ensure consistent application of the eligibility criteria. For every 650 references screened, they convened to resolve any discrepancies. The researchers converged on more than 95% of the screening decisions. A similar selection process was followed for full-text screening, where greater than 95% agreement was achieved as well.

The research team contacted the corresponding author of each study that was deemed eligible for inclusion in the review to inquire about new or unpublished papers relevant to the sustainability of WASH service delivery in schools. One paper included in the review was identified through this process.

### Data extraction and quality checks

The research team iteratively developed a data extraction template to collect information from all eligible studies. Two members of the team independently populated the template for each study in the sample. Information was extracted about the study population, research design, implemented intervention (if applicable), data-collection methods, and key findings. A third researcher compared the completed data extraction templates and identified discrepancies between the two. The three researchers met on a regular basis to discuss and resolve such discrepancies.

Quality appraisal checklists were adapted from the National Institute for Health and Care Excellence [21] and used to assess the risk for bias in all eligible experimental studies, quasi-experimental studies, and observational studies. Studies that included pre- and post-intervention measurement and randomly assigned treatment were classified as experimental; those with non-random assignment or only one round of data collection were considered quasi-experiments. Studies were classified as observational if they did not include a deliberate intervention. Studies were evaluated for risk of selection bias, attrition bias, reporting bias, and performance bias. Two researchers independently used the checklists to assess the risk for bias. A third researcher compared the completed quality appraisal checklists and resolved discrepancies between the two.

## Results

### Search results

The search across Web of Science, SCOPUS, and PubMed returned 3,841 unique studies. After full-text screening, 19 studies were included in the systematic review. The PRISMA flow diagram and reasons for exclusion are provided in Fig 1. Quality assessments of included studies can be found in S1–S6 Tables.

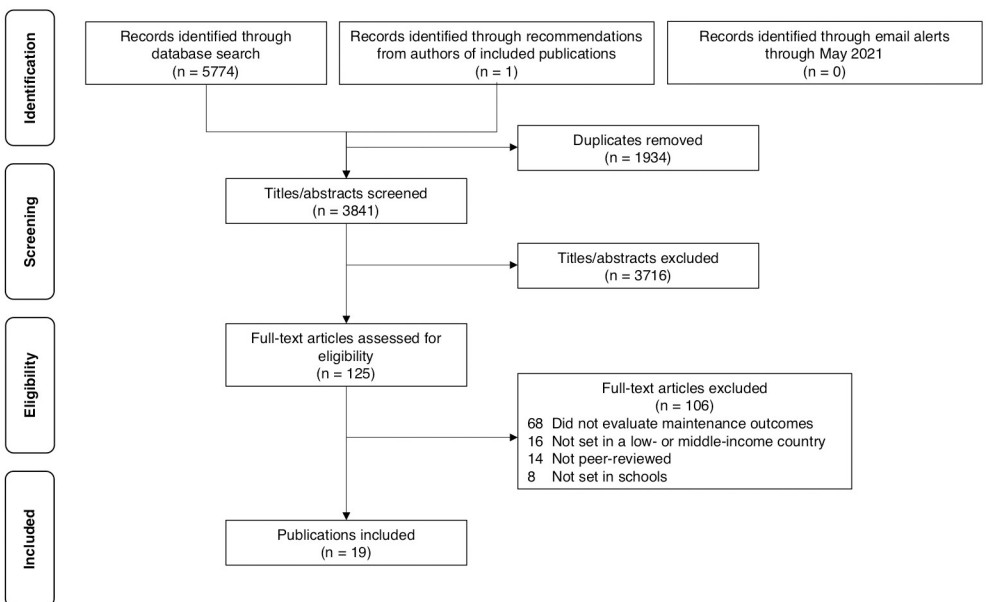

**Fig 1. PRISMA flow diagram detailing the number of studies included and excluded at each screening stage.**

## Observational studies

Observational studies included in our review shared the broad objective of identifying the enablers and barriers associated with sustainable WASH service delivery in schools. Most were initiated as *ex-post* investigations of school WASH programs that had delivered disappointing results [22–27]. Some observational studies were organized around a conceptual framework of hypothesized factors that support or inhibit sustainable service delivery. Others adopted more of a grounded theory approach in which the authors investigated pre-identified drivers while also allowing new ones to emerge [28]. The nine observational studies were conducted in Kenya (n = 3), Tanzania (n = 1), South Africa (n = 1), Nigeria (n = 1), Belize (n = 1), Bangladesh (n = 1), and Pakistan (n = 1) (Table 1).

All nine observational studies identified weak accountability and information sharing mechanisms as critical barriers to sustainable service delivery [22–27, 29–31]. We define accountability mechanisms as measures that confront individuals with meaningful consequences for their choices and behaviors. A functioning information sharing mechanism is one that delivers timely, credible information that individuals need for effective decision-making, such as their and others' roles, responsibilities, and consequences for failing to meet expectations. Clarity of roles is particularly critical when actors have shared responsibilities and when there is path dependency (i.e., when prior decision-making constrains options for future decision-making).

Some research teams underscored the importance of clearly communicated information, for example, about the roles and responsibilities of key actors in the service delivery system. Others emphasized the effective use of information in confronting these actors with meaningful consequences when they fail to fulfill their responsibilities. In Kenya, Snyder et al. (2020) found that strict organizational hierarchies prevented school administrators from communicating directly with the service providers. Hampering the flow of information in this way delayed corrective action when problems emerged with the schools' toilets [25]. In a separate study, researchers observed that schools without effective accountability mechanisms were less

**Table 1. Summary characteristics of observational studies that identified the enablers and barriers associated with sustainable WASH service delivery in schools.**

| Study | Country | Urban/ rural | Number of schools | Follow up of previously implemented intervention? | Outcome(s) of interest | Condition(s) of interest[a] | Data Collection Details |
|---|---|---|---|---|---|---|---|
| **Chatterley et al. (2013)** [22] | Belize | Rural and small towns | 15 | Yes | Functional toilets, with doors and locks, that are free of visible feces | [1] Local involvement upfront [2] Community support for operation and maintenance [3] Local champion [4] Vandalism [5] Quality of construction [6] Use of familiar technology | District Health and Family Life Education officers conducted unannounced school visits during the dry season. A checklist was used to systematically inspect facilities for repair needs, functionality, and cleanliness. Interviews were conducted in-person with principals and teachers. Students (aged 10–15 years) were interviewed at five schools when additional information was required. Two focus groups at the district level were conducted with community leaders, women's group representatives, and teachers. Focus groups incorporated both specific and open-ended questions. |
| **Chatterley et al. (2014)** [23] | Bangladesh | Rural | 16 | Yes | Functional toilets that are clean, accessible to students, and repaired in a timely manner | [1] Quality construction [2] Community support [3] Government support [4] Active school management committee [5] Maintenance plan [6] Sanitation champion | Unannounced visits were conducted at schools to collect information through interviews, focus groups, and systematic inspections of toilets. Separate semi-structured interviews were conducted with teachers and school-assigned field officers. Separate focus groups were conducted with four boys, four girls, and four "Little Doctors" (aged 9–11 years). Interviews and focus groups incorporated both specific and open-ended questions. |
| **Graves et al. (2013)** [24] | Kenya | Rural | 16 | Yes | Long-term implementation of primary school handwashing programs | [1] Teachers' expectations of health benefits (pre-implementation) [2] Teachers' observations of program benefits (post-implementation) [3] Teachers' observations of enablers of and barriers to program success [4] Teachers' perceptions of necessary resources required for program sustainability | Schools were monitored throughout the year by locally trained intervention staff members. Fourteen months after the initial implementation of the intervention, staff conducted structured interviews with a convenience sample of teachers from 16 schools. Up to three teachers were asked to participate from each school (a Head Teacher or Deputy Teacher, a teacher trained during the intervention, and a teacher not trained during the intervention). Each teacher was asked a standard series of open-ended questions. |
| **Ikoya, Peter (2008)** [29] | Nigeria | Urban, sub-urban, rural | 120 | No | Availability and functionality of physical facilities in schools | [1] School centralization or decentralization status | School administrators, teachers, students, and community members were interviewed. The interview collected information on respondent demographics, key variables related to the management of school facilities, and school management teams' participation in infrastructure management. School administrators provided records on student enrollment and physical facilities. These records were cross-checked by trained assistants during school visits. |

*(Continued)*

**Table 1.** (Continued)

| Study | Country | Urban/ rural | Number of schools | Follow up of previously implemented intervention? | Outcome(s) of interest | Condition(s) of interest[a] | Data Collection Details |
|---|---|---|---|---|---|---|---|
| **Mumtaz et al. (2019)** [30] | Pakistan | Urban, rural | 6 | No | Toilet cleanliness | [1] Design of WASH facilities [2] Cultural devaluation of toilet cleaners and inadequate governing practices | Data were collected from purposively sampled respondents through interviews with key informants (mothers, female teachers, health care providers, local religious leaders, and one scholar), participatory activities with girls aged 16–19 years (both in-school and out-of-school), observations of school infrastructure, and a document review of government policies, government websites, newspaper articles, and United Nations and NGO reports. |
| **Okello et al. (2019)** [27] | Tanzania | Urban, rural | 4 | Yes | Availability of handwashing water and soap | [1] Inconsistent availability of water [2] Inconsistent availability of soap at HW stations [3] Teachers and peers who support and encourage HWWS [4] Improved quality, quantity and location of HW stations [5] Proactive student engagement in water collection and soap provision | Two rounds of data collection, spaced six months apart, were conducted at four purposively selected schools. In the first round, researchers conducted two gender-segregated focus group discussions at each school with students aged 7–10 years. Four in-depth interviews with teachers were also conducted at each school. In the second round, five interviews with pairs of students and four in-depth teacher interviews were conducted at each school. |
| **Saboori et al. (2011)** [26] | Kenya | Rural | 55 | Yes | Provision of drinking water in safe water storage containers, treatment of drinking water, provision of handwashing water, and provision of soap near handwashing containers | [1] Financial capacity [2] Accountability [3] Technical feasibility and availability [4] Community support [5] School leadership and management [6] Student engagement | Data for this study were collected two and a half years after the completion of a pilot program across 55 schools. At each school, researchers conducted open-ended and structured interviews with either the head teacher or patron of the intervention. Researchers also conducted structured observations to evaluate (1) the conditions of water storage container, (2) the presence of water, and (3) the presence of soap. Lastly, stored drinking water was tested for residual chlorine. |
| **Snyder et al. (2020)** [25] | Kenya | Urban | 20 | Yes | Toilet accessibility, functionality, privacy, and cleanliness | [1] Reliability [2] Tangibles [3] Empathy [4] Responsiveness [5] Financial aspects [6] Assurance | Structured observations of sanitation facility conditions were conducted three times per year during a 3- to 4-year period at all study schools (n = 10). Key informant interviews were conducted with two administrative representatives at each school during the final year of this study. Head teachers, board of management members, champion teachers, and school staff with sanitation facility cleaning responsibilities were preferentially interviewed. |

(*Continued*)

**Table 1.** (Continued)

| Study | Country | Urban/rural | Number of schools | Follow up of previously implemented intervention? | Outcome(s) of interest | Condition(s) of interest[a] | Data Collection Details |
|---|---|---|---|---|---|---|---|
| **Xaba, Ike (2012)** [31] | South Africa | NR[b] | 16 | No | Level of coordination of maintenance activities in schools | [1] Maintenance organization<br>[2] Maintenance inspection<br>[3] Maintenance policies<br>[4] Maintenance planning<br>[5] Maintenance funding<br>[6] Service systems<br>[7] Maintenance categories | Respondents were purposively and conveniently selected for interviews. In total, 13 principals and 3 deputy principals from primary and secondary schools were interviewed to discuss the challenges associated with school facility maintenance. |

[a] Only explanatory factors related to the scope of this systematic review are included in this column.

[b] NR = not reported.

likely to consistently perform water treatment and latrine maintenance activities [26]. These authors pointed to the need for clear performance requirements, regular oversight of maintenance activities, and incentives to motivate compliance.

In Pakistan, Mumtaz et al. (2019) found that a lack of consequences for misconduct compromised an existing system that assigned cleaning staff to government schools [30]. The authors found that government-appointed cleaning staff were often tasked with cleaning the houses of the politically powerful, which left them no time to perform their contractual maintenance responsibilities at schools. Other staff who had been hired due to personal relationships with the local elite declined to carry out their toilet maintenance responsibilities, despite benefiting from a full salary [30]. This deprived schools of cleaning services altogether.

Ikoya (2008) found that those responsible for enforcing consequences often lack the authority to do so [29]. In Nigeria, decentralized schools with direct authority over their budgets had more functional facilities compared with centralized schools that relied on government agencies for maintenance support [29]. Ikoya (2008) attributed this difference to shortcomings in the existing accountability mechanisms; when teachers at centralized schools made requests for repair services, the Ministry of Education managed the process of selecting and dispatching contractors. Principals and school administrators were excluded from this process, and thus lacked the authority to hold contractors accountable if they procured low quality materials or executed tasks poorly [29].

Six of the nine observational studies identified reliable financial resources for operations and maintenance as a necessary component of sustainable WASH service delivery [22–26, 31]. However, many authors observed that information and accountability barriers persisted even when schools had adequate resources, resulting in unsustainable outcomes. For example, researchers in Bangladesh and Belize found that ongoing financial support from the government or the community was a necessary condition for clean and functional toilets, but it did not guarantee well-maintained facilities [22, 23]. In Bangladesh, one school experienced frequent toilet breakdowns despite having enough financial resources to repair the infrastructure [23]. The authors found that the teachers from this school were waiting for the implementing organization that built the toilets to take responsibility for fixing them, rather than initiating the repairs themselves.

Chatterley et al. (2013) found similarly misaligned expectations at several schools in Belize. Teachers interviewed by the research team noted that the responsibilities for ongoing operation and maintenance of school toilets were not clearly communicated to them by the Ministry of Education that originally constructed them. Teachers from these schools were still expecting

continued maintenance support from the government, asserting that "[The Ministry of Education] should come back to see it and make repairs" [22].

In Tanzania, Okello et al. (2019) found that soap availability at the handwashing station was inconsistent, even when soap was available at the school [27]. Teachers who were interviewed by the research team attributed the lack of soap to students' and teachers' neglecting to report its absence. The authors concluded that the inconsistent provision of soap was not a problem of resources but the result of weak information and accountability mechanisms.

## Experimental studies

The experimental (n = 6) and quasi-experimental (n = 4) studies included in the review were conducted in Kenya (n = 6), South Africa (n = 1), the Philippines (n = 1), Indonesia (n = 1), and India (n = 1) (Table 2). Experimental studies implemented interventions designed to increase the sustainability of existing WASH service delivery in schools. These interventions included multiple components, all of which primarily focused on the provision of resources in the form of cash disbursements, new infrastructure, and soap.

**Intervention components.** Nine of the ten experimental and quasi-experimental studies provided schools with cash payments or new WASH infrastructure [13–17, 19, 20, 32, 33]. In Kenya, schools received one-time financial disbursements to cover daily operational costs (0.44 USD per student), minor repairs (60 USD), and the salary of a part-time WASH attendant (120 USD, time period unspecified) [13]. In South Africa, plumbers visited intervention schools to perform water system maintenance and minor repairs [32]. These plumbers were given a list of authorized repairs and a fixed budget (5,000 Rand per school). In other studies, schools received new latrines [17, 19], latrine cleaning equipment (e.g., buckets, brooms, hand brushes, and plastic scoops) [15, 16, 20], and new handwashing facilities [14, 15].

In addition to financial resources and new infrastructure, schools in five of the ten studies were also provided with monitoring tools as part of the intervention package [13–16, 20]. In Indonesia, schools received new toilets and handwashing facilities, teacher training on monitoring and evaluation, and guidance to teachers and parents on developing a school action plan [14]. In Kenya, schools received buckets, brooms, hand brushes, and plastic scoops, in addition to latrine monitoring sheets for students to use daily [20]. In the Philippines, schools received handwashing infrastructure, toilet maintenance and cleaning tools, and monitoring sheets and checklists to be completed by school principals and staff [15].

Eight of the ten experimental and quasi-experimental studies included intervention components aimed at increasing the availability of WASH consumables such as handwashing soap and water [13–20]. Some research teams provided the consumables directly to schools [15–18, 20], whereas others provided schools with the financial resources to procure them [13]. A subset of studies also provided complementary monitoring tools for use by students, parents, and teachers [13, 15, 16, 20] and handwashing promotion and training materials [14, 16, 17, 19, 20].

**Impact of interventions in experimental studies.** Overall, the interventions described by studies included in this review had limited impact on WASH infrastructure maintenance in schools (Table 3). Of the five studies with statistically significant improvements in one or more outcome, four had mixed impacts [13, 14, 17, 19, 20]. For example, schools in Kenya that received cash transfers to cover daily operational costs, minor repairs, and the stipends of cleaning staff had significantly greater average latrine cleanliness scores (during unannounced spot checks over the four month follow-up period) than control schools [13]. The same interventions, however, did not measurably improve the observed functionality of drinking water taps (RR = 1, p = 0.78) or the number of latrines with a door (RR = 1, p = 0.43).

**Table 2. Summary characteristics of experimental and quasi-experimental studies that evaluated WASH infrastructure maintenance and consumables provision interventions in schools.**

| Study | Country | Urban/ rural | Intervention arms | Number of schools | Length of intervention | Timing of evaluation | Rounds of data collection |
|---|---|---|---|---|---|---|---|
| **Alexander et al. (2013)** [13] | Kenya | NR[a] | Treatment 1: Each school received a one-time disbursement of 0.44 USD per student. Treatment 2: In addition to the 0.44 USD per student, each school received monitoring tools for students and information on how to recruit a volunteer parent responsible for monitoring and reporting health issues to the school management committee. Treatment 3: In addition to 0.44 USD per student, each school received a one-time disbursement of 60 USD for minor repairs and an optional one-time disbursement of 120 USD to support a part-time WASH attendant. Control: No intervention. | Treatment 1: 15 Treatment 2: 15 Treatment 3: 15 Control: 25 | 2 months | Evaluation started in month 3 | 5 rounds (including baseline) over a period of 6 months |
| **Alexander et al. (2014)** [19] | Kenya | Rural | Treatment 1: Schools that reported receiving latrine construction or rehabilitation from a non-governmental organization (NGO). Treatment 2: Schools that reported receiving any water or handwashing interventions (e.g., handwashing promotion, handwashing materials, construction, or rehabilitation of water sources) from NGOs. Control 1: Schools that did not report receiving latrine construction or rehabilitation from an NGO. Control 2: Schools that did not report receiving water or handwashing interventions from an NGO. | Treatment 1: 43 Treatment 2: 32 Control: 15 | NA[b] | NA[b] | 1 round (cross-sectional) |
| **Alexander et al. (2018)** [18] | Kenya | Rural | Treatment 1: Schools received powdered detergent with instructions for making soapy water and a short instruction manual on basic WASH facility maintenance. Treatment 2: In addition to the powdered detergent and maintenance manual, study nurses distributed insertable, resuable bell-shaped menstrual containers (Mooncups®) to female students. Treatment 3: In addition to the powdered detergent and maintenance manual, study nurses distributed two packs of pads (Always®) per month. | Treatment 1: 10 Treatment 2: 10 Treatment 3: 10 | Not explicitly mentioned | Evaluation started between month 1–6 (not explicitly mentioned) | 6 rounds (including baseline) over a period of 17 months |
| **Bohnert et al. (2016)** [17] | Kenya | Urban | Treatment 1: Schools received 5 Fresh Life Toilets (FLTs), which are urine-diverting dry toilets with cartridges to collect the waste. A waste collection team removes and replaces the cartridge daily. Each school also received a hygiene curriculum to promote healthy WASH behavior change, a two-day training for teachers, one handwashing station (bucket with cover and tap), and waste cover material (sawdust). Treatment 2: Schools received a block of five cistern-flush toilets connected to the municipal septic system. One school received one block of five ventilated improve pit latrines instead due to a lack of construction space. Four schools also received rehabilitation of their existing toilet facilities. Each school also received between one and three handwashing stations (sink with multiple faucets). | Treatment 1: 10 Treatment 2: 10 | Treatment 1: 9–11 months[c] (median length: 11 months) Treatment 2: 3–5 months[c] (median length: 4 months) | Evaluation started in month 2 | 6 rounds (no baseline) over a period of 11 months |

*(Continued)*

**Table 2.** (*Continued*)

| Study | Country | Urban/rural | Intervention arms | Number of schools | Length of intervention | Timing of evaluation | Rounds of data collection |
|---|---|---|---|---|---|---|---|
| **Booysen, MJ (2019)** [32] | South Africa | Urban | Treatment: Plumbers performed basic maintenance and minor repairs of water systems at each school. Each plumber was given a checklist of permitted "quick fixes" and a total budget of Rand 5000. | Treatment: 196 | NA[d] | Evaluation started 1 day after intervention was implemented | 2 rounds (including baseline) over a period of 14 days[e] |
| **Buxton et al. (2019)** [15] | Philippines | NR[a] | Treatment: Schools received a detailed manual on toilet operation and maintenance (O&M) that included monitoring worksheets, budget allocation exercises, example cleaning rotas, checklists, and an O&M orientation video. These materials were developed based on existing monitoring requirements established by the national Department of Education, and designed to be used by school principals and staff. Each school also received a group handwashing facility, a toilet maintenance kit (including toilet brush, trash can, bucket and dipper), cleaning tools for each toilet, and a monthly supply of hygiene consumables (cleaning supplies, soap, toothpaste). Members of the intervention team also paid monthly visits to each school to provide guidance. Control: No intervention. | Treatment: 10 Control: 10 | 4 months | Evaluation started in month 4 | 2 rounds (including baseline) over a period of 4 months |
| **Caruso et al. (2014)** [20] | Kenya | Rural | Treatment 1: Schools received buckets, brooms, hand brushes, plastic scoops, bleach, toilet tissues, plastic bottles, and powdered soap. Health patrons (teachers that were responsible for the school WASH environment) and head teachers (supervisors of the health patrons and students) from each school were trained to instruct students to: (i) use the materials provided for latrine cleaning, (ii) monitor latrine conditions with a structured observation sheet, (iii) make soapy water, and (iv) wash hands at critical times. Schools were also provided with monitoring sheets designed to evaluate latrine conditions and supplies availability. Students were to use these sheets to monitor latrine conditions daily and supplies availability twice per week. Treatment 2: Schools received powdered soap, plastic bottles, and teacher training on (i) making soapy water, handwashing techniques, and critical handwashing times. Control: No intervention | Treatment 1: 20 Treatment 2: 20 Control: 20 | Not explicitly mentioned | Evaluation started in month 2 | 6 rounds (including baseline) over a period of 6 months |

(*Continued*)

**Table 2.** (Continued)

| Study | Country | Urban/ rural | Intervention arms | Number of schools | Length of intervention | Timing of evaluation | Rounds of data collection |
|---|---|---|---|---|---|---|---|
| **Saboori et al. (2013)** [16] | Kenya | Rural | Treatment 1: Schools received buckets, brooms, hand brushes, plastic scoops, bleach, toilet tissues, plastic bottles, and powdered soap. Health patrons (teachers that were responsible for the school WASH environment) and head teachers (supervisors of the health patrons and students) from each school were trained to instruct students to: (i) use the materials provided for latrine cleaning, (ii) monitor latrine conditions with a structured observation sheet, (iii) make soapy water, and (iv) wash hands at critical times. Schools were also provided with monitoring sheets designed to evaluate latrine conditions and supplies availability. Students were to use these sheets to monitor latrine conditions daily and supplies availability twice per week.<br>Treatment 2: Schools received powdered soap, plastic bottles, and teacher training on making soapy water, handwashing techniques, and critical handwashing times.<br>Control: No intervention | Treatment 1: 20<br>Treatment 2: 20<br>Control: 20 | Not explicitly mentioned | Evaluation started 2 weeks after intervention was implemented | 8 rounds (including baseline) over a period of 6 months |
| **Karon et al. (2017)** [14] | Indonesia | Rural | Treatment: Training was conducted for school committees, school representatives, and government and implementing partners in a WASH working group. The training covered a broad range of topics including operations and maintenance, monitoring and evaluation, the development of a school action plan among parents and teachers, and hygiene training on diarrhea, handwashing with soap, and clean drinking water. Schools received different interventions based on their respective action plans. In general, they received toilet and handwashing facility construction and water point rehabilitation.<br>Control: No intervention. | Treatment: 23<br>Control: 52 | 2 years[b] (2011–2013) | 1 year after intervention conclusion | 1 round (cross-sectional) |
| **Kochurani et al. (2009)** [33] | India | Urban, rural | Treatment[d]: The program provided schools with both hardware and software inputs that promoted handwashing, correct use of facilities, personal hygiene, and community outreach. School clubs were formed to clean facilities and classrooms, conduct community meetings, and deliver classes on personal hygiene, safe drinking water, and sanitation.<br>Control: No intervention. | Treatment: 150<br>Control: 150 | 1–5 years[b] (1999–2003) | 3–8 years after intervention conclusion (2006–2007) | 1 round (cross-sectional) |

[a] NR = not reported.

[b] Quasi-experimental study that did not implement or assign the intervention.

[c] Varying intervention lengths due to implementation complications.

[d] The intervention was still active when authors were preparing the manuscript.

[e] Data collection was continuous via sensors.

Some studies that included infrastructure and resources for maintenance as part of their intervention documented no measurable impact. For example, schools in the Philippines were provided with handwashing facilities, toilet cleaning tools, and monitoring worksheets, budget allocation exercises, and cleaning checklists for use by school principals and staff [15]. Four months later, intervention schools had not significantly improved relative to control schools with respect to observed toilet accessibility (RR = 0.9, p = 0.74), functionality (RR = 0.94, p = 0.27), quality (RR = 1.07, p = 0.66), or overall usability (RR = 0.86, p = 0.64) [15].

Interventions that included a consumables provision component had mixed impacts on the availability of supplies (Table 3). Of the eight studies, five found a statistically significant increase in the availability of soap at handwashing stations [13, 14, 16, 18, 20] and three found a statistically significant increase in the availability of handwashing water [13, 14, 19]. Whereas the availability of soap significantly increased in several studies, it was never available all the

**Table 3. Summary of infrastructure maintenance and consumables provision outcomes reported by experimental and quasi-experimental studies.**

| Study | Data[a] | Infrastructure maintenance outcomes | | | | | Consumables provision outcomes | | | |
|---|---|---|---|---|---|---|---|---|---|---|
| | | Functionality of drinking water facilities | Functionality of handwashing facilities | Accessibility of latrines | Structural integrity and functionality of latrines | Latrine cleanliness | Drinking water | Handwashing water | Soap | Latrine cleaning supplies |
| Alexander et al. (2013) [13] | O | green | green | | green | | green | green | green | |
| | O | green | green | | yellow | | green | green | green | |
| | O | green | green | | green | | green | green | green | |
| Alexander et al. (2014) [19] | O | | | | yellow | yellow | | | | |
| | O | | | | | green | | green | black | black |
| | R | | | | | | | yellow | black | black |
| Alexander et al. (2018) [18] | O | | | | red | red | | yellow | green | black |
| | R | | | | | | | yellow | green | yellow |
| Bohnert et al. (2016) [17] | O | | | yellow | | green | | | | black |
| | R | | | | | yellow | | | | |
| Booysen, MJ (2019) [32] | O | black | | | | | | | | |
| Buxton et al. (2019) [15] | R | | | yellow | yellow | yellow | | | | |
| Caruso et al. (2014) [20] | O | | | | | green | | yellow | green | |
| | O | | | | | yellow | | yellow | green | |
| Saboori et al. (2013) [16] | O | | | | | | | yellow | green | |
| | R | | | | | | | yellow | green | |
| | O | | | | | | | yellow | green | |
| | R | | | | | | | green | green | |
| Karon et al. (2017) [14] | O | yellow | | black | yellow | green | | green | green | |
| Kochurani et al. (2009) [33] | R | | | | black | black | black | | black | |

[a] O = observed, R = reported.

[b] Cells shaded in green indicate a significant improvement (p ≤ 0.05); cells shaded in yellow indicate a non-significant impact (p > 0.05); cells shaded in red indicate a significant deterioration (p ≤ 0.05); cells shaded in black indicate that no statistical significance was reported; and cells shaded in light grey indicate that the outcome was not measured.

[c] Some assumptions have been made for brevity. See S7–S9 Tables for indicator definitions and S10 and S11 Tables for exact p-values.

time. In a study with one of the largest measured impacts, intervention schools were statistically more likely, on average, to have soap available during unannounced spot checks compared to control schools ($p < 0.001$) [13]. Though these intervention schools were provided with a combination of cash transfers, monitoring worksheets, and information on how to recruit parents to contribute to regular monitoring at schools, soap was still only available for less than half of the time (42–50%) [13].

## Discussion

The literature included in this systematic review suggests that most efforts to improve the sustainability of WASH service delivery in schools have failed. Interventions focused on improving the maintenance of infrastructure have had limited impact on the accessibility, cleanliness, and structural integrity of latrines. Interventions focused on improving the regular provision of consumables have increased the availability of soap but have had limited impact on the availability of water for handwashing. These conclusions hold true both for interventions that delivered financial resources alone and for multicomponent interventions that provided monetary and infrastructural resources along with training and monitoring support.

The disappointing performance of school WASH interventions is less surprising when considered in tandem with insights that emerge from observational studies included in the review (Fig 2). Authors of these studies identified a lack of accurate and timely information about maintenance needs and procedures, weak enforcement of existing policies and responsibilities, and limited authority of school personnel to make maintenance-related decisions as key drivers of unsustainable WASH services. These are not elements of the WASH service delivery system targeted by most experimental research, however. Instead, interventions have largely focused on enhancing the availability of resources, from cash and infrastructure to cleaning supplies and consumables.

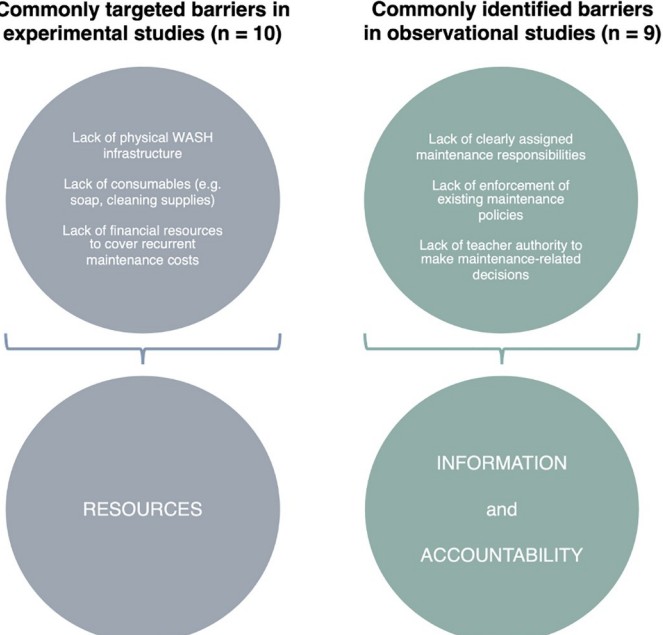

**Fig 2. Commonly targeted barriers in experimental studies contrasted with commonly identified barriers in observational studies.**

One explanation for the difference in emphasis is that many of the observational studies were *ex-post* evaluations of resource-enhancing interventions that yielded disappointing results. Despite efforts to provide schools with key resources, their WASH facilities remained poorly maintained and consumables continued to be unavailable. In seeking explanations for the less-than-anticipated impact of these resource-enhancing interventions, observational studies revealed the importance of ensuring information flow and accountability mechanisms—areas not originally targeted by the interventions—in addition to addressing resource constraints.

Similar learnings are reflected in experimental studies. In most cases, the experimental studies (n = 5) that intervened to help engage teachers, students, and parents in regular WASH monitoring focused on the provision of tools and teacher trainings; they did not delve into the ways that monitoring data might be used to enhance accountability. Caruso et al. (2014) and Saboori et al. (2013) acknowledge this limitation in their approaches [16, 20]. Both authors provided schools with worksheets and trainings to engage students and teachers in regular latrine and consumables monitoring. The interventions primarily focused on ensuring teacher and student comprehension of how and when to use the monitoring tools. There is no evidence that the interventions included incentives for correct and consistent use of the tools. Both authors acknowledged their limited exploration of, and the need to understand more thoroughly, ways in which students and teachers could be motivated to carry out their monitoring responsibilities.

In Kenya, Alexander et al. (2013) provided schools either with a single cash transfer of 0.44 USD/student, or with the funding plus a set of WASH service monitoring supports. Students were given monitoring tools and trained on their use, and a volunteer parent representative was recruited to collect and report information on WASH conditions to the school management committee (SMC) [13]. The authors documented no significant difference in impact between the two interventions. They did note that, whereas all intervention schools established the volunteer parent representative position and completed the student monitoring forms, the level of engagement among parents and students varied greatly across schools. Parent volunteers in particular were often too busy to fulfill their monitoring responsibilities, and they were not confronted with meaningful consequences for failing to play their assigned role.

Providing monitoring tools and trainings without concomitant attention to incentivizing their use is a recurring theme among the experimental studies included in the review. In the Philippines, Buxton et al. (2019) provided schools with a detailed manual on toilet O&M that included monitoring worksheets, budgeting exercises, sample cleaning rotas, and checklists to be completed by school principals and staff [15]. The resources were developed to support schools' compliance with the Department of Education monitoring requirements already in place. The tool-focused intervention, however, had no measurable effect on the share of schools that fulfilled their monitoring obligations. Nor did it differentially enhance the accessibility, functionality, and quality of toilet facilities, even in schools that used the tools regularly.

As with other studies in this group, Buxton et al. do not present evidence that a lack of tools, or knowledge about how to use them, was the key impediment to regular monitoring and maintenance of school toilets. Indeed, their findings suggest that a more important obstacle may be whether and how WASH service data are linked to professional incentives for school personnel. Several experimental researchers did acknowledge the role that information and accountability play in a well-functioning WASH service system. Some mentioned the low uptake of monitoring tools by teachers [15], limited prioritization of WASH responsibilities among school staff [13, 18, 19, 32], and lack of clarity around WASH responsibilities within the school environment [19]. There is less reflection on whether the provision of funding, tools, teacher trainings, monitoring worksheets, and often uncompensated responsibilities to

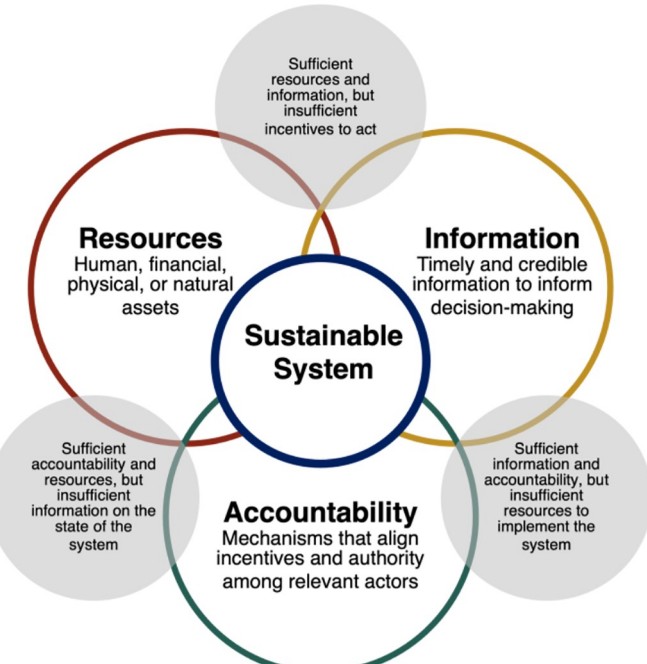

**Fig 3. Resources, information, and accountability are each necessary, but insufficient components of sustainable service delivery systems.**

members of the school community will provide sufficient motivation to overcome these barriers.

The evidence generated by these studies gives rise to the hypothesis that adequate resources, timely and credible information, and functioning accountability mechanisms together constitute the necessary components of a sustainable WASH service delivery system (Fig 3). None of the studies we reviewed intervened in all three domains (Fig 4). Interventions that target accountability mechanisms are particularly demanding to develop and, not surprisingly, relatively less common among the studies we reviewed. They require skill sets and disciplinary perspectives that have not featured prominently in WASH research to date, along with local expertise regarding sociocultural and institutional norms. At the same time, the results of this review suggest that continued focus on resource-only interventions is unlikely to yield meaningful gains in sustainable WASH service for schools. Unless and until interventions also address information needs, and the formal and informal incentive structures that shape individual behaviors, they will likely fall short of achieving sustainable impact.

Going forward, research that identifies the types of information bottlenecks that regularly impede sustainable WASH service delivery in schools would be a valuable contribution. Testing cost-effective strategies for delivering timely, accurate information could also be useful. Information and communications technologies hold promise here, particularly for more objective information about, for example, the status of WASH facilities and supplies. Several studies have highlighted the potential of sensor technologies to provide real-time information about service quality and use, and to improve service delivery [34–38]. These technologies can facilitate more responsive or even proactive service delivery, but only if they generate information that decision-makers consider relevant and credible, and that they are incentivized to consider [39].

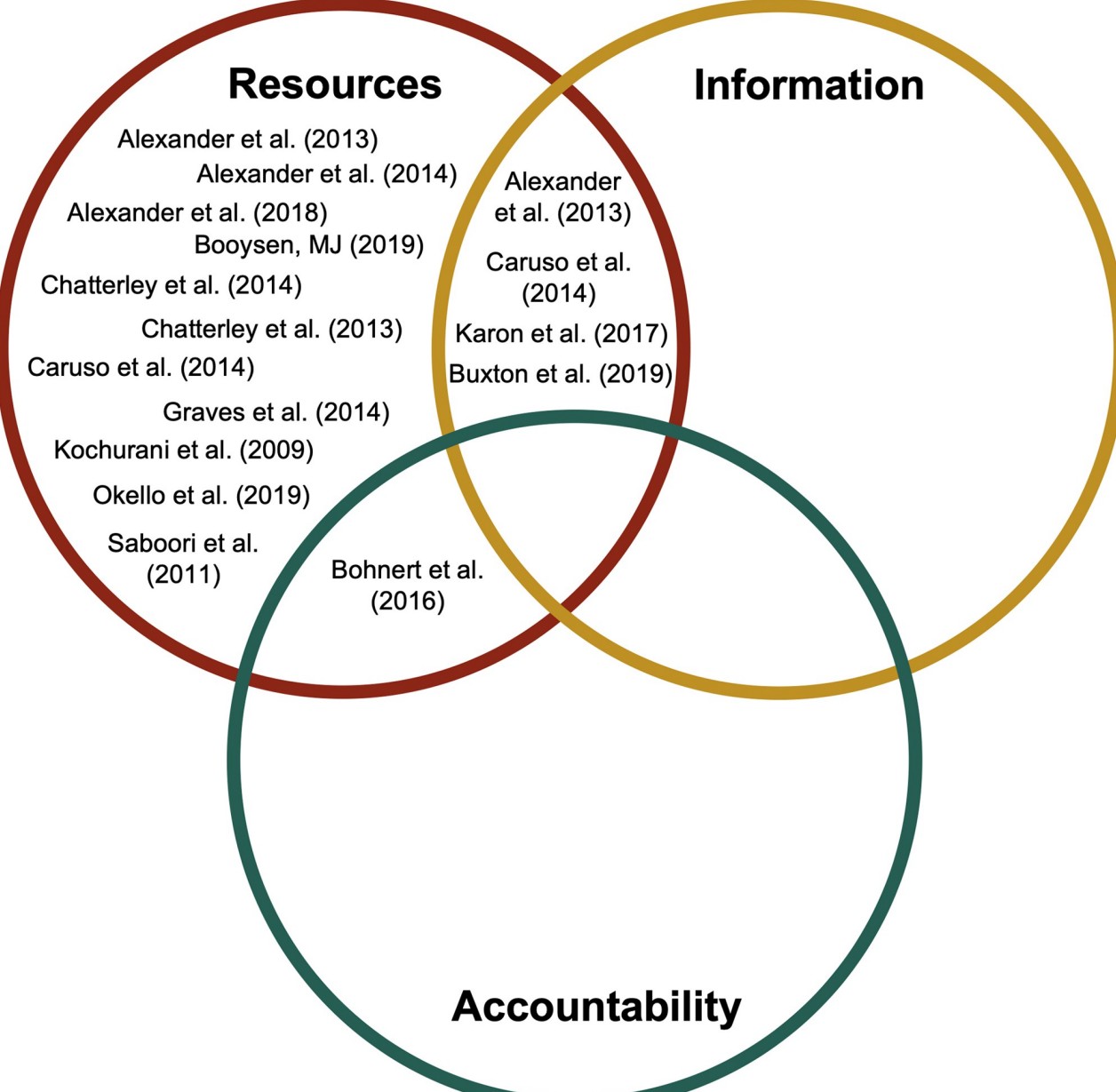

**Fig 4. Classification of experimental, quasi-experimental, and observational studies based on components of implemented interventions.** Studies may appear more than once if they featured multiple intervention arms. Saboori et al. (2013) and Snyder et al. (2020) are not included because the same interventions are already represented by Caruso et al. (2014) and Bohnert et al. (2016), respectively.

Other strategies to facilitate the flow of information—in particular regarding the needs, expectations, experience, and perceptions of individuals—include public expenditure tracking surveys, complaint procedures, community monitoring programs, citizen report cards, public hearings, and social audits [40]. As with sensor-based technologies, these approaches can improve WASH service delivery only if the information they produce is integrated effectively into decision-making processes and is used to present meaningful consequences to people. Recent research by Kumpel et al. (2020) is instructive in this regard. The authors developed

information flow diagrams to map formal and informal systems for managing information within water quality monitoring programs across six countries in sub-Saharan Africa [41]. They identified several barriers to effective information sharing, including the limited aggregation, analysis, and sharing of data by water suppliers, surveillance institutions, and regulatory agencies.

Thomas and Brown (2021) emphasize the need to align financial, political, and social incentives to increase the likelihood that the information generated will be acted upon. Their work illustrates how timely and actionable monitoring and decision support mechanisms can be designed for different types of stakeholders, including recipient communities, program implementers and service providers, researchers, and donors [42]. Similar efforts in other contexts would yield a better understanding of key service delivery design elements, such as (1) priority information needs of key actors, (2) effective information distribution channels, and (3) strategies to link information flows with accountability mechanisms.

Literature on experimental and behavioral research into accountability suggests that whether an accountability mechanism functions effectively depends on four key attributes: (1) timing (i.e., whether an individual expects to be held accountable before or after decision-making), (2) source (i.e., the party/ies to whom one feels accountable), (3) salience (i.e., the extent to which individuals are held accountable for outcomes that they perceive to be important for achieving the overall goal), and (4) standard (i.e., whether individuals are held responsible for processes or outcomes) [43–45]. Evidence suggests that expecting to be held accountable for a decision prior to making it [43], being held accountable to someone in a respected position of authority [44], and believing that one's actions will have significance [45] increase the time and effort an individual spends on making an informed decision.

The literature on accountability standards is more mixed [46]. Advocates of process accountability contend that if well-planned processes are emphasized, favourable outcomes will naturally follow [45]. Critics argue that process accountability can encourage blind conformity to inadequate procedures [46]. By contrast, outcomes accountability can compensate for inadequate processes by facilitating innovation and creativity. However, critics caution that outcomes accountability may penalize individuals for things beyond their control [47], overemphasize poorly understood metrics, and create an environment ripe for shortcuts and unethical behavior [45, 46].

In light of these considerations, it would be fruitful to explore alternative institutional arrangements for WASH service delivery in schools. Common approaches to service delivery rely on volunteers, both as individual "champions" and members of committees who lack the resources, authority, political support, and incentives to hold important stakeholders accountable [48]. Those expected to demand accountability from service providers often have the least agency to do so [49].

Contracting out maintenance responsibilities may present advantages here. Bohnert et al. (2016) advocate for schools to have the option of outsourcing service delivery responsibilities to private sector vendors, citing lower financial costs and higher quality services. Outside the school setting, contracting out WASH maintenance responsibilities has been associated with greater system functionality, faster repair time, lower rates of microbial contamination, and improved satisfaction among households [50]. Whether outsourcing is successful, however, depends on the complexity and risks associated with the service, as well as the governance model and organizational capacity of the delegator [51–53]. Moving forward, it would be productive to assess the conditions under which different service delivery models are able to establish effective information flows and accountability mechanisms, and how these compare with respect to economic and political feasibility.

Our study is not the first to propose a framework for sustainable service delivery, but it does deviate others in two ways. First, existing frameworks tend to be complex; the list of factors relevant for consideration is diverse and exhaustive. Absent guidance on what to prioritize, such frameworks may be of limited utility to decision makers with scarce resources and bandwidth. By contrast, our framework is parsimonious. We synthesized the important causal conditions identified by studies in this review into three system components that we hypothesize are simultaneously necessary. Our hope is that this provides a more manageable, flexible and, ultimately, useful approach to designing and implementing interventions with sustainable impact.

Second, existing frameworks and recommendations stipulating where and how to intervene are often context dependent; their applicability across differing circumstances is poorly understood. This framework can be used to investigate the underlying causes of unsustainable service provision in complex systems with unique political and socio-cultural environments. The process is systematic but not generic: local expertise is essential to operationalize the concepts of resources, information, and accountability in ways that inform the design of effective and context-sensitive interventions. It embodies the notion of "best fit" approaches rather than overly simplistic "best practice" thinking [54].

Indeed, whereas we focused this systematic review on WASH services in schools we note that the tendency to pursue resource-focused interventions to enhance service delivery has been documented in other contexts and sectors. Findings in this review reflect conclusions drawn about the sustainability of smallholder irrigation systems [55], health services [56], as well as human-environment systems at large [57]. This evidence suggests that the resources-information-accountability framework might be usefully employed to analyses of other systems as well.

We do acknowledge the limitation of restricting this review to peer-reviewed literature published in English. Professional or "grey" literature and work published in other languages may contain important insights that are not captured in this review. The lack of geographic diversity in the current body of literature may also limit the generalizability of our findings. We hope that studies on sustainable WASH service delivery, in schools and in other settings, will continue to expand beyond its current focus in Kenya. Our goal was to pursue disaggregated analyses along different school dimensions (e.g., urban *versus* rural, public *versus* private). Unfortunately, the studies in this review were insufficiently diverse along these dimensions; authors also rarely provided disaggregated data to facilitate such comparisons. Finally, the diversity and complexity of the interventions tested in these studies makes it difficult to assess the effects of particular intervention elements, much less to directly compare interventions across studies. We also note that formal meta-analysis was precluded by the use of non-standardized outcome indicators across a range of time scales.

Infrastructure systems directly or indirectly influence 72% of the Sustainable Development Goals' targets, and investments in infrastructure will continue to be essential for delivering basic services [58, 59]. As evidenced by the studies reviewed here, however, physical assets are a necessary but insufficient condition for reliable service delivery. Key knowledge gaps remain regarding the types of investment that support locally effective information and accountability mechanisms, and the conditions under which their integration with resource provision leads to truly sustainable service delivery.

## Supporting information

**S1 Table. Quality assessments of observational studies.**
(PDF)

**S2 Table. Quality assessment rubric for observational studies.**
(PDF)

**S3 Table. Quality assessment of quasi-experimental studies.**
(PDF)

**S4 Table. Quality assessment rubric for quasi-experimental studies.**
(PDF)

**S5 Table. Quality assessment of experimental studies.**
(PDF)

**S6 Table. Quality assessment rubric for experimental studies.**
(PDF)

**S7 Table. Indicator definitions and data collection details associated with experimental and quasi-experimental studies that evaluated maintenance outcomes pertaining to water facilities.**
(PDF)

**S8 Table. Indicator definitions and data collection details associated with experimental and quasi-experimental studies that evaluated maintenance outcomes pertaining to hand-washing facilities.**
(PDF)

**S9 Table. Indicator definitions and data collection details associated with experimental and quasi-experimental studies that evaluated maintenance outcomes pertaining to sanitation facilities.**
(PDF)

**S10 Table. Reported outcome statistics from experimental and quasi-experimental studies that implemented interventions with infrastructure maintenance components.**
(PDF)

**S11 Table. Reported outcome statistics from experimental and quasi-experimental studies that implemented interventions with consumables provision components.**
(PDF)

**S1 Protocol. Registered PROSPERO systematic review protocol.**
(PDF)

**S1 Checklist. PRISMA 2020 reporting guidelines for systematic reviews.**
(PDF)

## Acknowledgments

We thank Anja Zehfuss for her contributions to the data collection process.

## Author Contributions

**Conceptualization:** Christine JiaRui Pu, Gary L. Darmstadt, Jennifer Davis.

**Data curation:** Christine JiaRui Pu, Poojan Patel, Gracie Hornsby.

**Formal analysis:** Christine JiaRui Pu, Gary L. Darmstadt, Jennifer Davis.

**Funding acquisition:** Christine JiaRui Pu, Gary L. Darmstadt, Jennifer Davis.

**Investigation:** Christine JiaRui Pu, Poojan Patel, Gracie Hornsby.

**Methodology:** Christine JiaRui Pu, Gary L. Darmstadt, Jennifer Davis.

**Project administration:** Christine JiaRui Pu.

**Supervision:** Christine JiaRui Pu, Gary L. Darmstadt, Jennifer Davis.

**Validation:** Christine JiaRui Pu, Poojan Patel, Gracie Hornsby.

**Visualization:** Christine JiaRui Pu, Poojan Patel, Gracie Hornsby.

**Writing – original draft:** Christine JiaRui Pu.

**Writing – review & editing:** Christine JiaRui Pu, Poojan Patel, Gracie Hornsby, Gary L. Darmstadt, Jennifer Davis.

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
