## [Decision Letter · Decision Letter 0]

1 Apr 2022

PONE-D-22-06487Necessary conditions for sustainable water and sanitation service delivery in schools: A systematic reviewPLOS ONE

Dear Dr. Pu,

Thank you for submitting your manuscript to PLOS ONE. After careful consideration, we feel that it has merit but does not fully meet PLOS ONE’s publication criteria as it currently stands. Therefore, we invite you to submit a revised version of the manuscript that addresses the points raised during the review process.

The three reviewers have provided some varied comments which I would like you to address.

We look forward to receiving your revised manuscript.

Kind regards,

Alison Parker

Academic Editor

PLOS ONE

Journal Requirements:

3. We note that this manuscript is a systematic review or meta-analysis; our author guidelines therefore require that you use PRISMA guidance to help improve reporting quality of this type of study. Please upload copies of the completed PRISMA checklist as Supporting Information with a file name “PRISMA checklist”.

Reviewers' comments:

Reviewer's Responses to Questions

**Comments to the Author**

1. Is the manuscript technically sound, and do the data support the conclusions?

Reviewer #1: Partly

Reviewer #2: Yes

Reviewer #3: Yes

2. Has the statistical analysis been performed appropriately and rigorously? 

Reviewer #1: N/A

Reviewer #2: Yes

Reviewer #3: Yes

3. Have the authors made all data underlying the findings in their manuscript fully available?

Reviewer #1: Yes

Reviewer #2: Yes

Reviewer #3: Yes

4. Is the manuscript presented in an intelligible fashion and written in standard English?

Reviewer #1: Yes

Reviewer #2: Yes

Reviewer #3: Yes

5. Review Comments to the Author

Reviewer #1: overall this is a well designed study and a clear presentation of the background and results. My primary concerns related to the discussion and synthesis of the findings.

Specifically this sentence: In short, experimental studies have largely intervened by enhancing the availability of resources, whereas observational studies found that information and accountability mechanisms are in need of greater attention.

The authors need to provide a better definition of what constitutes an intervention directly addressing information and accountability in order to justify this statement. By my own count, Alexander (2013); Buxton, Caruso, Saboori and Karon all address institutional accountability and information in some way - which are collectively over half of the interventional studies included. An argument could be made that the interventions by Bonhert and Booysen both address this by outsourcing these responsibilities. More specificity is needed in the results to properly differentiate why the authors do not consider these in line with the results from observational studies.

The manuscript currently lacks details on the limitations of the current review and how these limitations may have influenced the results. A few examples include focusing solely on peer-reviewed literature rather than the wealth of grey literature that dominates much of the knowledge base for school-WASH interventions, challenges in defining interventions and their specific components, etc.

Suggested edit - the authors can decide if this is worthwhile: if the main point is to underscore the differences between observational and "interventional" studies, perhaps starting with the observational studies would better underscore the disconnect between the focus of the two?

Reviewer #2: This is a good article which identifies some of the key conditions for sustainable WASH service delivery. It would have been good to separate schools depending on their different settings. For example, there are often major disparaties between schools in rural and urban settings. In addition, some schools are private while others are public. Making such distinctions could have made the article much more interesiting particularly with regards to availability of resources(rural vs schools) and accountability (private vs public schools).

Reviewer #3: This is a concise, well-written paper that should be of widespread interest in the WASH community.

I have no major comments or concerns with this manuscript. I urge the authors to consider calling for greater funding for school-based WASH in settings of interest for this review. Although the authors conclude that "The results of this review suggest that continuing to focus on resource-only

interventions is unlikely to meaningfully improve the sustainability of WASH service

delivery in schools." (lines starting with 383), it is true that, generally, schools operate without proper funding in LMICs, and a policy case can be made that greater public funding for schools is going to be required to fix the issues highlighted by this review. Schools can be an important vehicle for infrastructure and health progress in communities.

Material beginning on line 405: this paper has covered very similar material on the utility of accountability, and may be of interest: https://pubs.acs.org/doi/10.1021/acs.est.0c04115

6. PLOS authors have the option to publish the peer review history of their article (what does this mean?). If published, this will include your full peer review and any attached files.

Reviewer #1: No

Reviewer #2: No

Reviewer #3: No

---

## [Author Response · Author response to Decision Letter 0]

19 May 2022

Journal Requirements

Thank you for this reminder. We have ensured that our manuscript meets PLOS ONE’s style requirements. 

We have ensured that our reference list is complete and correct. We have not cited any papers that have been retracted. We have made the following changes to the reference list:

1. Added reference [2] in the introduction section

2. Added reference [42] in the discussion section, as recommended by reviewer 3

3. We note that this manuscript is a systematic review or meta-analysis; our author guidelines therefore require that you use PRISMA guidance to help improve reporting quality of this type of study. Please upload copies of the completed PRISMA checklist as Supporting Information with a file name “PRISMA checklist”.

We have uploaded a copy of the completed PRISMA checklist under the file name “S7_PRISMA checklist.pdf” in the compressed “Supporting Information.zip” file.

Reviewers’ comments

Reviewer #1

Overall this is a well designed study and a clear presentation of the background and results. My primary concerns related to the discussion and synthesis of the findings.

Specifically this sentence: In short, experimental studies have largely intervened by enhancing the availability of resources, whereas observational studies found that information and accountability mechanisms are in need of greater attention.

The authors need to provide a better definition of what constitutes an intervention directly addressing information and accountability in order to justify this statement. By my own count, Alexander (2013); Buxton, Caruso, Saboori and Karon all address institutional accountability and information in some way - which are collectively over half of the interventional studies included. An argument could be made that the interventions by Bonhert and Booysen both address this by outsourcing these responsibilities. More specificity is needed in the results to properly differentiate why the authors do not consider these in line with the results from observational studies.

We thank the reviewer for the encouraging and constructive feedback. We have incorporated definitions for these terms in the results section on lines 192 - 198, as follows:

“We define accountability mechanisms as measures that confront individuals with meaningful consequences for their choices and behaviors. A functioning information sharing mechanism is one that delivers timely, credible information that individuals need for effective decision-making, such as their and others’ roles, responsibilities, and consequences for failing to meet expectations. Clarity of roles is particularly critical when actors have shared responsibilities and when there is path dependency (i.e., when prior decision-making constrains options for future decision-making).”

We reviewed our classifications of the experimental and quasi-experimental studies vis-a-vis the resource, information, and accountability categories. We agree with the reviewer that Alexander et al. (2014), Buxton et al. (2019), Caruso et al. (2014) and Saboori et al. (2014) included intervention components that aimed to improve monitoring / information flows. We have updated Table 2 and Figure 4 to more clearly communicate these intervention components. We have also elaborated on these research teams’ information intervention components on lines 285 - 293, as follows:

“In addition to financial resources and new infrastructure, schools in five of the ten studies were also provided with monitoring tools as part of the intervention package [13–16,20]. In Indonesia, schools received new toilets and handwashing facilities, teacher training on monitoring and evaluation, and guidance to teachers and parents on developing a school action plan [14]. In Kenya, schools received buckets, brooms, hand brushes, and plastic scoops, in addition to latrine monitoring sheets for students to use daily [20]. In the Philippines, schools received handwashing infrastructure, toilet maintenance and cleaning tools, and monitoring sheets and checklists to be completed by school principals and staff [15].”

Even with these re-classifications, it is still the case that most of the interventions developed and tested experimentally by authors focused on increasing the financial and material resources available to schools for WASH service delivery. Only one study interviewed explicitly on accountability issues (Bohnert et al.). We have tried to tighten up our presentation of this finding throughout the manuscript, such as the text we modified on lines 361-370:

“The disappointing performance of school WASH interventions is less surprising when considered in tandem with insights that emerge from observational studies included in the review. Authors of these studies identified a lack of accurate and timely information about maintenance needs and procedures, weak enforcement of existing policies and responsibilities, and limited authority of school personnel to make maintenance-related decisions as key drivers of unsustainable WASH services. These are not elements of the WASH service delivery system targeted by most experimental research, however. Instead, interventions have largely focused on enhancing the availability of resources, from cash and infrastructure to cleaning supplies and consumables.”

The manuscript currently lacks details on the limitations of the current review and how these limitations may have influenced the results. A few examples include focusing solely on peer-reviewed literature rather than the wealth of grey literature that dominates much of the knowledge base for school-WASH interventions, challenges in defining interventions and their specific components, etc.

We thank this reviewer for pointing out this important omission. We have added a discussion of the limitations of this review on lines 569 – 583:

“We do acknowledge the limitation of restricting this review to peer-reviewed literature published in English. Professional or “grey” literature and work published in other languages may contain important insights that are not captured in this review. The lack of geographic diversity in the current body of literature may also limit the generalizability of our findings. We hope that studies on sustainable WASH service delivery, in schools and in other settings, will continue to expand beyond its current focus in Kenya. Our goal was to pursue disaggregated analyses along different school dimensions (e.g., urban versus rural, public versus private). Unfortunately, the studies in this review were insufficiently diverse along these dimensions; authors also rarely provided disaggregated data to facilitate such comparisons. Finally, the diversity and complexity of the interventions tested in these studies makes it difficult to assess the effects of particular intervention elements, much less to directly compare interventions across studies. We also note that formal meta-analysis was precluded by the use of non-standardized outcome indicators across a range of time scales.”

Suggested edit - the authors can decide if this is worthwhile: if the main point is to underscore the differences between observational and "interventional" studies, perhaps starting with the observational studies would better underscore the disconnect between the focus of the two?

We thank the reviewer for this terrific idea. We have flipped the order of presentation as suggested. Leading with the observational studies does indeed lay out the disconnect between observational versus experimental studies more effectively. 

Reviewer #2

This is a good article which identifies some of the key conditions for sustainable WASH service delivery. It would have been good to separate schools depending on their different settings. For example, there are often major disparaties between schools in rural and urban settings. In addition, some schools are private while others are public. Making such distinctions could have made the article much more interesiting particularly with regards to availability of resources(rural vs schools) and accountability (private vs public schools).

We thank this reviewer for this feedback. We agree that disaggregated analyses along these dimensions (public vs. private, urban vs. rural) would provide interesting layers of depth. Our intention was to explore and elaborate on these differences as well. However the set of studies we reviewed (1) were insufficiently diverse with respect to these dimensions, and (2) rarely provided disaggregated data along these dimensions to facilitate trend analyses. Without sufficient sample sizes and information, we are unable to draw conclusions about the important disparities the reviewer raised. We have, however, added a brief discussion of this limitation in the manuscript, on lines 575 – 578:

“Our goal was to pursue disaggregated analyses along different school dimensions (e.g., urban versus rural, public versus private). Unfortunately, the studies in this review were insufficiently diverse along these dimensions; authors also rarely provided disaggregated data to facilitate such comparisons.”

Reviewer #3: This is a concise, well-written paper that should be of widespread interest in the WASH community.

I have no major comments or concerns with this manuscript. I urge the authors to consider calling for greater funding for school-based WASH in settings of interest for this review. Although the authors conclude that "The results of this review suggest that continuing to focus on resource-only interventions is unlikely to meaningfully improve the sustainability of WASH service delivery in schools." (lines starting with 383), it is true that, generally, schools operate without proper funding in LMICs, and a policy case can be made that greater public funding for schools is going to be required to fix the issues highlighted by this review. Schools can be an important vehicle for infrastructure and health progress in communities.

We thank this reviewer for the encouraging feedback. To be sure, we are not saying that schools do not need (more) resources to deliver sustainable WASH services. We are saying that investing in infrastructure and consumables can fail to deliver impact if the information and accountability dimensions of sustainable service delivery are lacking. We have revised the text on lines 585 – 592 to convey this message more clearly:

“Infrastructure systems directly or indirectly influence 72% of the Sustainable Development Goals’ targets, and investments in infrastructure will continue to be essential for delivering basic services [58,59]. As evidenced by the studies reviewed here, however, physical assets are a necessary but insufficient condition for reliable service delivery. Key knowledge gaps remain regarding the types of investment that support locally effective information and accountability mechanisms, and the conditions under which their integration with resource provision leads to truly sustainable service delivery.”

Material beginning on line 405: this paper has covered very similar material on the utility of accountability, and may be of interest: https://pubs.acs.org/doi/10.1021/acs.est.0c04115

We appreciate the reviewer for pointing us to this resource. We agree this reference contains complementary material, and have elaborated on its contents in lines 487 – 495:

“Thomas and Brown (2021) emphasize the need to align financial, political, and social incentives to increase the likelihood that the information generated will be acted upon. Their work illustrates how timely and actionable monitoring and decision support mechanisms can be designed for different types of stakeholders, including recipient communities, program implementers and service providers, researchers, and donors [42]. Similar efforts in other contexts would yield a better understanding of key service delivery design elements, such as (1) priority information needs of key actors, (2) effective information distribution channels, and (3) strategies to link information flows with accountability mechanisms.”

---

## [Decision Letter · Decision Letter 1]

20 Jun 2022

Necessary conditions for sustainable water and sanitation service delivery in schools: A systematic review

PONE-D-22-06487R1

Dear Dr. Pu,

We’re pleased to inform you that your manuscript has been judged scientifically suitable for publication and will be formally accepted for publication once it meets all outstanding technical requirements.

Kind regards,

Alison Parker

Academic Editor

PLOS ONE

Additional Editor Comments (optional):

Reviewers' comments:

Reviewer's Responses to Questions

**Comments to the Author**

1. If the authors have adequately addressed your comments raised in a previous round of review and you feel that this manuscript is now acceptable for publication, you may indicate that here to bypass the “Comments to the Author” section, enter your conflict of interest statement in the “Confidential to Editor” section, and submit your "Accept" recommendation.

Reviewer #1: All comments have been addressed

Reviewer #2: All comments have been addressed

2. Is the manuscript technically sound, and do the data support the conclusions?

Reviewer #1: Yes

Reviewer #2: Yes

3. Has the statistical analysis been performed appropriately and rigorously? 

Reviewer #1: N/A

Reviewer #2: Yes

4. Have the authors made all data underlying the findings in their manuscript fully available?

Reviewer #1: Yes

Reviewer #2: Yes

5. Is the manuscript presented in an intelligible fashion and written in standard English?

Reviewer #1: Yes

Reviewer #2: Yes

6. Review Comments to the Author

Reviewer #1: Thank you for thoughtful consideration of comments. I look forward to seeing the final published version.

Reviewer #2: I think this is a good piece of work which can influence policy and practice. The WASH sector is a complex space with so many actors and it requires strong policies, institutions, legislations that can provide the enabling environment. It is important to identify the necessary conditions that can sustain WASH services and the authors have done a good job of highlighting these conditions.

7. PLOS authors have the option to publish the peer review history of their article (what does this mean?). If published, this will include your full peer review and any attached files.

Reviewer #1: No

Reviewer #2: No

---

## [Editor Report · Acceptance letter]

24 Jun 2022

PONE-D-22-06487R1 

Necessary conditions for sustainable water and sanitation service delivery in schools: A systematic review 

Dear Dr. Pu:

I'm pleased to inform you that your manuscript has been deemed suitable for publication in PLOS ONE. Congratulations! Your manuscript is now with our production department. 

Kind regards, 

on behalf of

Dr. Alison Parker 

Academic Editor

PLOS ONE